# Healthcare providers as patients: COVID-19 experience

**Abbas Al Mutair**[1,2,3,4,5]*, Alexander Woodman[6], Amal I. Al Hassawi[7], Zainab Ambani[7], Mohammed I. Al Bazroun[7], Fatimah S. Alahmed[7], Mary A. Defensor[7], Chandni Saha[1], Faiza Aljarameez[8]

**1** Research Center, Almoosa Specialist Hospital, Al-ahsa, Saudi Arabia, **2** School of Nursing, University of Wollongong, Wollongong, Australia, **3** Almoosa College of Health Sciences, Al-ahsa, Saudi Arabia, **4** Nursing Department, Prince Sultan Military College, Dhahran, Saudi Arabia, **5** Department of Medical-Surgical Nursing, Princess Nourah Bent Abdulrahman University, Riyadh, Saudi Arabia, **6** School of Health Sciences, University of Salford, Manchester, United Kingdom, **7** Nursing Department, Qatif Central Hospital, Qatif, Saudi Arabia, **8** King Saud bin Abdulaziz for Health Sciences, King Abdullah International Medical Research Center, Ministry of the National Guard Health Affairs, Al-Hasa, Saudi Arabia

* abbas.almutair@almoosahospital.com.sa

**Data Availability Statement:** There is an ethical restriction against sharing the minimal dataset imposed by the Qatif Central Hospital IRB. The reason is to protect research participants data. A non-author point of contact that is able to receive

## Abstract

There is compelling evidence for the psychological effects of the COVID-19 pandemic and earlier epidemics. However, fewer studies have examined the subjective meaning experience of healthcare providers who have survived COVID-19 as patients. This qualitative study aimed to understand further and describe the life experiences of healthcare providers who have survived COVID-19 as patients in Saudi Arabia. Data was collected using unstructured in-depth individual interviews among n = 10 healthcare providers from public hospitals in Saudi Arabia. Data were analyzed based on a phenomenological approach, which resulted in five themes: (i) physical and psychological signs and symptoms; (ii) self-healing, hiding pain, and family; (iii) fear of complications; (iv) disease stigma & long-term psychological outcomes; (v) emotional support, mental well-being & resignation. The overall synthesis showed that healthcare providers, as patients, experience the same difficulties and stressors as the general public. In some cases, these factors are even worse, as family members, colleagues, and employers develop a new type of stigma. Given the impact of social media and the flow of information of any type, more research is needed to examine the sources used to obtain information by the general public, whether these sources are reliable, and how the public can be taught to use only scientific data and not social data. Understanding the experience of healthcare providers as patients during the pandemic has allowed to look at the feelings and needs of people during illness from a new perspective. As expressed by participants, being a healthcare provider does not reduce the fear of the disease and does not mitigate its consequences in the form of stigmatization and isolation.

## Introduction

The experience of the past two years has shown that COVID-19 has affected the global community both physically and psychologically [1–3]. The emergence of the COVID-19 pandemic was associated with massive changes in all aspects of society. This has impacted the healthcare

queries regarding data access is er-qch-research@moh.gov.sa.

**Funding:** The author(s) received no specific funding for this work.

**Competing interests:** The authors have declared that no competing interests exist.

system worldwide as huge global efforts have been directed toward combating the COVID-19 pandemic. Much of this effort has focused on developing therapeutic and preventive medical measures to treat those affected [4–6].

The scientific literature has been enormously enriched with data, experiences, and success stories from the global medical community and the general [1, 7, 8]. However, studies mainly focused on the physiological aspects of COVID-19, and fewer looked into the psychological outcomes for those affected by the virus. Available evidence suggests that the disease affects not only the physical condition but also the mental health and well-being of the individual and society, leaving the individual with long-term consequences [6, 9]. During COVID-19, in addition to the stresses during the disease, various factors such as disruption to the daily routine, lockdowns, and long-term health conditions after recovery from the virus significantly and profoundly affected the mental health of the global community [5, 6, 10].

A study in northern Italy found that 17.2% of COVID-19 survivors had post-traumatic stress disorder (PTSD) symptoms four months after being discharged from the hospital [4]. An earlier study by Maunder et al. (2003) exploring patients' experiences during the SARS outbreak in Toronto and found fear, loneliness, boredom, and anger [11]. In contrast, a qualitative study conducted among Spanish patients showed that patients easily adapted to the situation and trusted medical professionals [12]. The isolation of patients was perceived as a necessity, and technology helped keep patients and their families connected. Furthermore, a qualitative study among Turkish survivors found denial and adaptation, fear, and hopelessness associated with patients' psychological health [13].

Taking into consideration patients' experience, healthcare professionals should be mentally resilient, trained, and experienced in dealing with patients' health condition, death, and mental health [3, 14]. However, stress and the unique demands of the COVID-19 crisis put healthcare providers at additional risk of mental health issues, indicating increased levels of depression, anxiety, and post-traumatic stress disorder. A systematic review of 46 quality papers by Billings et al. (2021) sought to better understand the experiences of frontline healthcare providers and their perspectives on support during COVID-19 and previous epidemics [3]. Meta-synthesis has identified that participants in all studies were deeply concerned about their own physical safety and/or the physical safety of others. This was the largest in the early stages of the pandemic and was exacerbated by inadequate personal protective equipment (PPE), insufficient resources, and conflicting information. In addition, workers struggled with high workloads and long shifts and wanted adequate rest and recovery [3].

A similar systematic review among healthcare providers in the Kingdom of Saudi Arabia, aimed at assessing their mental health outcomes while coping with the COVID-19 crisis, found that more than 50% of healthcare providers experienced a psychological impact in terms of depression, anxiety, and stress during the global pandemic emergency [2]. As such, the earlier study by Almutairi et al. (2018), studying the experiences of healthcare providers who survived MERS-CoV, reported four main themes related to the psychological impact of the disease [15]. These were caring for others in the defining moments, perceived prejudice behaviors and stigmatization, lived moments of traumatic fear and despair, and denial and underestimation of the seriousness of the disease at the individual and organizational levels. The survivors suffered from the traumatic experience, which could have had a negative impact on their productivity [2, 15].

Despite the existence of a number of works on the main topics related to the psychological consequences of the COVID-19 pandemic and earlier epidemics, there is a lack of research on the subjective meaning experience among healthcare providers who survived the COVID-19. Therefore, this qualitative study aimed to further understand and describe the life experiences of healthcare providers infected with COVID-19 in Saudi Arabia.

## Methods

### Design & sample

The present cross-sectional study applied qualitative design using a phenomenological approach, i.e., how a phenomenon (the lived experience of COVID-19) is experienced at the time it occurs. The target sample of participants who met the inclusion criteria were health professionals aged 18 years and older who agreed to be interviewed.

Purposive sampling was utilized in this study. Purposive sampling is a technique that allows researchers to identify and select participants who went through the phenomenon of interest and can enrich the study with their perspectives. Recruitment of participants was performed based on pre-defined inclusion criteria and it continued until data saturation was reached. This study was conducted with the participation of n = 10 healthcare providers from public hospitals in Saudi Arabia to explore the different views and opinions of medical professionals working in various medical institutions.

### Procedures

Ethical approval to conduct the study was obtained from the Qatif Central Hospital Scientific Research Ethics (QCH-SRECO207/2020). Participants provided their signed consent to participate in the interview, and their responses were anonymized throughout the study to maintain confidentiality. Participants' names were coded during analysis to ensure privacy. The in-depth interviews were conducted by three bilingual interviewers with experience in the healthcare system. The interviews were audio- recorded and transcribed manually. The collected data was stored in a password-protected device accessible only to the research team.

### Data collection

Given the exploratory nature of the study, data collection was carried out using unstructured in-depth individual interviews. The flow of interviews was from the general to the specific. The interviewer started asking general questions, such as the participant's demographic information, then moved toward the core questions of the topic. This interview sequence, however, was different for all participants as it depended on the interview process and each participant's responses [16]. The main position n of the interviewer was to encourage participants to talk freely about the topics in the interview guide. This approach allowed participants to talk about what they thought was important [17].

### Data analysis

The thematic analysis aims to achieve an understanding of patterns and meaning based on the experience of participants (e.g., descriptions of experiences related to the research question by healthcare providers) [18]. The analysis started with the transcription of the data, which is required to organize the meanings found in the data into patterns and themes. By reading the textual data several times and referring to the aim of the study, familiarity with the data was achieved. This allowed to immerse in the text and its meanings, to explore the experiences expressed in the data, narratives expressed and how the meanings can be understood. This stage allowed highlighting new patterns and reflect on what is already known [18].

This stage was followed by the search for meanings. Moving back and forth between the entire text and its parts contributed to a deeper understanding of the narrative [18]. Expressions corresponding to the aim of the study were marked in addition to the annotations so that preliminary themes could be identified. Meanings related to each other were compared to reveal differences and similarities to gain insight into patterns. This stage was followed by the

organization of meanings into patterns and finally into themes. In this phase, the meanings and themes derived from the original data were constantly compared by reflecting on the preliminary themes emerging from the data [18]. Findings were written and rewritten to allow faithful descriptions of meanings. Finally, the meanings derived from the participants' experiences were described through themes [18].

## Validity and rigor

Although there is no consensus on what concepts should be used regarding validity in qualitative and phenomenological research, specific strategies were maintained to ensure rigor and validity through reflexivity, credibility, and transferability. In this study, reflexivity was maintained throughout the process, which included questioning the understanding of the findings and themes rather than taking them for granted. As a result, the collected data was constantly compared to the descriptive text of the themes. In addition, the results were supported by participants' thoughts to demonstrate how derived descriptions are based on data rather than the researcher's understanding. As the credibility focuses on the significance of findings and whether they are well presented, the researcher sought to emphasize how the analysis and conclusions are presented to the reader, to be transparent, and to strive for validity. The themes described have been illustrated with quotations to ensure consistent content and meanings described. Through transferability, this study aimed to find out whether the results are valid and whether the study adds new knowledge to what is already known. In particular, the relevance, usefulness, and significance of the study results in other contexts were important to consider to transfer the study [18–20].

## Results

Ten healthcare providers participated in this study; six were Saudis, three were Philippines, and one was Jordanian. Eight participants were nurses and two food servers. The age of participants ranged from 26 and 39 years, with an average of 33.4 years. The participants worked in secondary hospitals, in medical, surgical, and Operating Theater departments, with an experience of 10 years and above.

The descriptive thematic analysis resulted in five themes, and four sub-themes (see Table 1) from participants lived experiences:

**Table 1. Themes and sub-themes from the qualitative analysis of the lived experiences of COVID-19 survivors in Saudi Arabia.**

| Theme | Subtheme | Code | Example |
|---|---|---|---|
| **Theme 1:** Physical and psychological signs and symptoms | | Participants talked about the signs and symptoms they experiences | Example 1: "*heavy flu symptoms, loss of smell and taste.*"<br>Example 2: "*I felt like I was a victim of physical abuse. Aside from the worsening cough and colds, headache, and fever, the pain all over my body was excruciating.*"<br>Example 3: "*I was shocked when I could no longer taste my food. I couldn't even smell it. . .*"<br>Example 4: "*I did (have illusions), I imagined a lot of things in the hospital, seeing shapes and ghosts. . .*"<br>Example 5: "*I lost the desire to talk to anyone*" |
| **Theme 2:** Self-healing, hiding pain, and family | | Participants described how they handle the situation | Example 1: "*I'm telling myself to be positive and strong through the help of God, I know I can surpass this kind of situation. . .*"<br>Example 2: "*I have an eldest daughter, she was about to cry when she knew that I am infected, I was about too, but I hold myself to not scare them.*"<br>Example 3: "*When I see them, I always prepare myself, I put on makeup, lipstick especially when I face my husband. I don't want him to see that I look pale so he will not worry about me*".<br>Example 4: "*Because they are the ones who get weak and affected, that's why the patient is the one who is giving strength so they will see that everything will go back to normal.*" |

*(Continued)*

**Table 1.** (Continued)

| Theme | Subtheme | Code | Example |
|---|---|---|---|
| **Theme 3:** Fear of complications | Self and family's reaction and fear from death | Patients Describing their fears | Example 1: "...none of the family yet knew or saw any actual positive cases... So COVID-19 is considered by them as alarming news even considered as immediate death."<br>Example 2: "My oldest daughter, aged 10 years old when I told her that I was positive and that her dad wasn't, so he would stay with them, she started to cry..."<br>Example 3: "I was shocked, unbelievable! Oh my God, why? It was like heaven and earth dropped on me."<br>Example 4: "I was not hugging my kids, I stopped since the pandemic happened" |
| | External interference in the face of breaking news | | Example 1: "...since everything on the news is negative that's why probably sometimes when people hear that you are positive with the virus as if they will think everything is going to be bad."<br>Example 2: "They talked to me with horror but still supportive... Because of the social media all that comes to their mind is COVID-19 and ventilators."<br>Example 3: "I went to their room and told them, they were shocked and my little girl got scared." |
| **Theme 4:** Disease stigma & long-term psychological outcomes | | Patients describing the immediate stigma faced & psychological impact | Example 1: "...the food was left on the door outside, I am understanding that they are afraid of being infected, they are not coming to the room..."<br>Example 2: "As soon as I arrived, the first thing I saw was the Police car, I did not like that, because I felt like a criminal..."<br>Example 3:"No one wanted to come close to me... you (the nurse) are already wearing a face mask, what will you do!"<br>Example 4: "I wasn't worried about myself rather worried about my kids and the staff who I was with" |
| **Theme 5:** Emotional support, mental well-being & resignation | The vital role of emotional support in healing | Patients describing how they received support during this period | Example 1: "Colleagues are also giving me moral support by calling and saying that "I know you can do it". They are making me strong and for me not to lose hope, that "don't worry, Allah will bless you..."<br>Example 2: All of them are here for me, supporting me ...even if we don't have family here, at least your colleagues, your friends are really there to give support."<br>Example 3: "I was so thankful because their kindness helped me feel better" |
| | Work resignation | | Example 1""As soon as I finished explaining, my mom asked me to resign immediately"<br>Example 2: I really felt so down and I couldn't stop crying<br>Example 3: I was so depressed. Instead of getting better, I became weaker. And at that moment, I decided that I will listen to my mother. I decided it's time for me to resign." |

1. **Physical and psychological signs and symptoms**

2. **Self-healing, hiding pain, and family**

3. **Fear of complications**

   a. Self and family's reaction and fear from death

   b. External interference in the face of breaking news

4. **Disease stigma & long-term psychological outcomes**

5. **Emotional support, mental well-being & resignation**

   a. The vital role of emotional support in healing

   b. Work resignation

## Physical and psychological signs and symptoms

Most of the symptoms and signs reported by the participants were common. These include severe flu symptoms, loss of smell and taste, difficulty getting out of bed, increased coughs and colds, headache, and fever.

> "*. . . heavy flu symptoms, loss of smell and taste.*"

> "*I have difficulty getting out of bed because I felt like I was a victim of physical abuse. Aside from the worsening cough and colds, headache, and fever, the pain all over my body was excruciating. Even taking a bath was so difficult because I felt like my skin was about to be lacerated when touched. I was forcing (myself) to eat a little just so I can take medications. I wanted to cry but I was pushing myself to be strong because, at this time, I can only depend on myself.*"

Participants also associated the loss of their sense of taste and smell with shock, which led to mind games and hesitations about whether it was the first day or whether the loss of taste and smell happened the day before or even earlier. In some cases, participants lost the desire to talk or interact with others.

> "*On that same day at lunch time, I was shocked when I could no longer taste my food. I couldn't even smell it. I took my perfume to smell it but nothing. I started asking myself. Did it really start today? Or it was already there from the beginning but because I constantly remind myself that I am negative that I didn't notice it? Did my mind play a trick on me during the previous days? That I was just imagining the taste of my food because I keep telling myself that my result will never become positive?*"

> "*I lost the desire to talk to anyone. . . I felt like tired, exhausted, I don't want to talk to anyone even if it's only to say "hello". . . sometimes I can't sleep at all to the point I cry for help asking for pills*".

Some cases were severe and required admission to the intensive care unit and intubation. Of these cases, some participants reported that they could hardly hear and see things during intubation, but they had hallucinations, and often the forms of the people seemed to be ghosts or dark shadows. This resulted in less sleep due to fear of new hallucinations.

> "*Yes, (I was able to hear suctioning sound and staff). I remember one voice. The voice, I assume, of the respiratory therapist. I do not know his name, I used to hear him talking to me. Whenever they do the ABG from time to time, I see them. When I woke up, I felt the needles and I heard sounds like suction*"; "*I did (have illusions), I imagined a lot of things in the hospital, seeing shapes and ghosts. I told myself that it is the death angel. I saw a lot of things to the point I could not sleep so most of my time after, After I woke up, most of the time, I could not go back to sleep. I felt tired, I imagined black shapes around me*".

## Self-healing, hiding pain, and family

Participants consciously or unconsciously practiced self-healing, i.e., treating their physical and/or mental problems to feel better. They explained this by fears for their health and, most importantly, for relatives and friends. They were worried about how their family members would see and feel about their health condition. Hence, they pretended to feel or look good, always laughing or putting on an extra layer of makeup to avoid looking tired or pale.

"*I'm telling myself to be positive and strong through the help of God, I know I can surpass this kind of situation, and then I will just pray. That's why I told myself and my family not to worry about me because I don't have any symptoms.*"

"*When I see them, I always prepare myself, I put on makeup, lipstick especially when I face my husband. I don't want him to see that I look pale so he will not worry about me". "when he called, he was already crying. Of course, when you are reading something about people getting positive with COVID, they were thinking something else. . .That a lot of people are dying, there are really lots of people who became critical, that's why that was his first reaction and he did not stop from crying that's why I told him not to cry because I'm OK. I don't have any symptoms, I'm strong.*"

They further explained that such an approach is necessary because family members and relatives often experience even more stress than the patients themselves, especially children of infected parents. In addition, another level of responsibility was added by the fact that the participants were medical professionals, which implies a greater responsibility towards the individual and their psychological well-being.

"*Because they are the ones who get weak and affected, that's why the patient is the one who is giving strength so they will see that everything will go back to normal.*"

"*I have an eldest daughter, she was about to cry when she knew that I am infected, I was about too, but I hold myself to not scare them.*"

"*I am crying here with the girls (nurses) whenever I see my father but I never cried in front of my family. I tried to look stronger in front of them because they may say she's a medical staff and she's crying which means she saw something unbearable, so I tried strengthening myself in front of them then to come here and cry it out. I tried my best to hide it, to not show it at all to them.*"

## Fear of complications

### Self and family's reaction and fear from death

Participants reported that being away from their home countries and the daily reports of high numbers of deaths made them feel even more stress and anxiety, both for themselves and, most importantly, for family members who were away. Even worse were the psychological consequences for those participants who were parents and had to isolate themselves from their children for some time. It was not easy to explain the reasons for isolation to children under ten.

"*Both of my parents, honestly were disturbed, our relatives as well, as you know my family is an expat. I am an expatriate! . . . none of the family yet knew or saw any actual positive cases. . . So COVID-19 is considered by them as alarming news even considered as immediate death.*"

"*My oldest daughter, aged 10 years old when I told her that I was positive and that her dad wasn't, so he would stay with them, she started to cry. I kept reassuring her by telling her that everyone got it and they were just doing fine and that I would be talking to them, behind the door.*"

"*I was not hugging my kids, I stopped since the pandemic happened, poor kids. They even, sometimes break my heart. I mean, I have to wash myself before I touch them. Since the*

*pandemic happened, I was strict, with hand hygiene all the time even my hands from the hygiene started to burn. . .They asked me why did I do so? I told them that it is for the best. They didn't understand why I'm doing it."*

Some health care workers who monitored COVID-19 patients and their complications every day were also shocked because they did not know how to break the news to family or colleagues, what their reaction would be.

*"I was shocked, unbelievable! Oh my God, why? It was like heaven and earth dropped on me. The first thing that came to my mind was how am I going to tell my family. How will I tell them that I am positive? How. . .how about my colleagues? I was thinking that what if they will feel gross about me. We are seeing how they are treating patients with COVID, and how they are trying to get distant from them. I'm worried that they will treat me the same way, they will feel disgusted with me. . ."*

In most cases, stress and feelings of shock were associated with the belief that the attending physician should provide care instead of being treated, leading to a nervous breakdown in the participants, fear of death or isolation, and long-term outcomes.

*"I was just stressed on the 1st day in the hospital because I did not expect it that it will happen to me. Like, with all the people, why this thing happened to me, why only me? I was just thinking about how I will explain it to my family. I was also scared of how my colleagues will treat me. Probably they will not come to me or talk to me since I got this virus."*

*"Positive. I wanted to ignore the fear but I couldn't. I started crying. I can't stop my tears. Thoughts started filling my mind. All the fears that I was trying to ignore started to consume me. I watch the news every day and thousands are dying because of this disease. Even those healthy individuals, with no co-morbidities who were infected also succumbed to death. Will I be like them? What will happen to me? I am alone in my room. Who will know if I will still be alive tomorrow or in the days to come? Will I tell my family of my positive result? Questions after questions started playing in my head that I wasn't able to sleep that night."*

Families of patients who saw and experienced disease complications experienced the same experience when the patients were infected. Thus, the participants noted that another task was to reassure their family members and give them hope that everything was in order. In addition, some family members had nightmares after visiting patients in the hospital, which had a long-term effect on their mental health. Therefore, participants often considered it their duty to comfort loved ones, despite the severity of their physical and psychological health at the time of infection.

*"They told me how it was painful on them, how terrible the situation was, especially my sister who came to visit me in the ICU. . .Till now, she is having nightmares, she is the one having nightmares not me. She told me that because she saw me and saw how terrifying my situation was that is why she is still having nightmares, till now she sees those things like black shapes and so on and she cannot sleep well."*

*"When it comes to my family, of course, they are so worried because I'm alone here and isolated. I was pushing them not to get stressed. I told them that if they will not stop worrying, I will worry about them instead because they are the ones who will get sick. I told them, "It's important that you should think that I am not sick, I am okay". "Don't think that I am sick,*

*that I am alone here because I have lots of friends here who take care of me, assisting me". "I don't want them to always ask about my case. . .I feel sad because I'm thinking that my family always thinks of my sickness. It makes me worried that later on, that's the only thing they will remember, that I am COVID positive like what others think. If you become positive, there's no chance for you to live, that this virus is scary. . . It really matters that when we talk to them, they see that you are happy so they will not get stressed."*

## External interference in the face of breaking news

Participants reflected on the negative impact of social media and external distractions, which often negatively impacted both the infected and the doctors.

*". . .since everything on the news is negative that's why probably sometimes when people hear that you are positive with the virus as if they will think everything is going to be bad."*

*"They talked to me with horror but still supportive. . . Because of the social media all that comes to their mind is COVID-19 and ventilators."*

External factors exacerbated the efforts of many participants to share a positive test result, especially with their family members. It was particularly stressful when breaking the news to parents who were elderly and unclear about how they would take it, which was accompanied by fear of transmission before the test.

*"I was blank face because, at that time, I was thinking how will I explain this to my family? how am I going to tell them?"*

*"I told my mother that I would not come over neither the girls, she was shocked, I was afraid that even my girls, they could be infected and they might infect my family. My mother asked me whether I am truly assuming that it is a COVID-19 or not. I told her that I am. She was shocked and immediately she became silent."*

*"My family, aunts, and uncles, they could've died, May Allah forbid it. I mean when they knew that it was COVID-19, they were frightened of the name in itself so when they knew, they were pretty worried."*

*"I went to their room and told them, they were shocked and my little girl got scared and started to cry. I reassured her that I am okay. They think whoever gets infected dies". "I told them that I'm okay as you can see me. My middle girl she's really a worrying person she immediately feels sick "Maam, I have fever and diarrhea" She made me worry even more. . .I don't know if these are symptoms or only out of fear. When your beloved one fears something you wouldn't let it go, you want to make sure."*

## Disease stigma & long-term psychological outcomes

When hospitals no longer had the space or staff to care for large numbers of COVID-19 patients, they often transferred infected patients to designated hotels for quarantine until recovery. Some patients felt more comfortable and relaxed in these hotels, while others felt social stigma and how, in some cases, they were treated like criminals who could escape without police protection. In addition, stigmatization was reflected in the service and care at the hotel, as well as the staff's fear of infection.

*"As soon as I arrived, the first thing I saw was the Police car, I did not like that, because I felt like a criminal, all that is left is to handcuff me. It was not a nice scenario, maybe, some people may try to escape, but I do not think so. . . It was not right, as if I was a criminal. They guided me to the waiting area, we sat there till they finished the process, and they took the ID then went to the Nurses' station."*

*"Now I moved to the next stage which is the hotel. There, of course, they specified rooms for patients "there is your room" and so on. . . the food was left on the door outside, I am understanding that they are afraid of being infected, they are not coming to the room or whatever but forgive me for the word, but you feel like impure! You have to give them the trash can, leaving it on the door to change it for you even walking in the corridor can criminalize you, the bright side is the quietness but the instruction is a bit, I don't know. Of course, they will not come to you but whatever you need, they will put it on the door for you so it was okay."*

The way people look at and treat infected patients was a real stress for some of the participants, while others did not experience any backlash from colleagues or the community.

*"I did not go through situations with anyone, everyone knew that I have COVID-19. In the end it is not something I need to hide. Honestly no one made me feel as if "oh, she has COVID-19. . . Wow. . ." or anything of that sort, it was normal."*

*"It's really a consuming thing for the mind. . . I mean, from a social point of view and people. How do they look at a person who is infected with this thing as if he is not sick, as if she is the one who brings shame. I mean, in my whole life whether the social or my work life, there's no (such reaction to) disease that spread. . . I mean, the ways of transmission are known. They are not, may Allah forbid it, through forbidden things or things that are not good or something. I mean the feeling, look or words that are spreading, as if this person is a shame, it is a crime, it is a forbidden thing."*

Some participants felt that people were avoiding and running away from them, even colleagues and closest friends as if they were the source of the infection. Even during treatment, other healthcare providers were reluctant to come closer. Although participants considered it somewhat normal, the facial expressions and energy emanating from friends and medical professionals were shocking.

*"I went out of the room down to the elevator and when they saw me as if they saw a beast. You would never imagine what happened even if I stood now and imitate what they have done, they all took a step behind and lift their hands away, far away, stay there. That was on the next day when my mental health was down. Just imagine how your feelings are at that moment. I don't know how to cry; I don't know how to be silent. I mean, I sat like a piece of wood. They saw a chair then they brought it to me, they told me to sit on it. I don't know how to express such. . . sadness".*

*"No one wanted to come close to me. . . you (the nurse) are already wearing a face mask, what will you do!"*

*"They were also shocked but I understood if nobody came near me, nobody comforted me but I know from their voice that they were worried and concerned. I felt what they were saying it's just that they cannot come near me physically in particular but the words they said comforted me."*

Social stigma and people's attitudes led to feelings of guilt in some participants. In addition, those who worked in high-risk areas and had daily contact with patients fear becoming carriers of the disease among family members and colleagues.

"*Psychologically, things were very difficult because even my wife is pregnant, and I began to think about her a lot, what could happen to her as I am the reason behind her infection.*"

"*I wasn't worried about myself rather worried about my kids and the staff who I was with, so I was daily asking the group if the result came out... Alhamdulillah, they were all negative... If anyone of them were positive, I could have died. I feel guilty for, what did I do wrong*?! *Why did I get infected in the first place*?"

## Emotional support, mental well-being & resignation

### The vital role of emotional support in healing

All participants emphasized the importance of emotional support in healing and overcoming the infection. Messages from family members or friends greatly impacted patients whenever they felt overwhelmed.

"*Colleagues are also giving me moral support by calling and saying that "I know you can do it". They are making me strong and for me not to lose hope, that "don't worry, Allah will bless you...", I'm being uplifted, there are lot of people who believe in me that makes me strong...a lot of people are saying that they are praying for me, "you are included in our prayers that's why don't lose hope...It added to my self-confidence that soon everything will be arranged and hopefully, my next swab will become negative".*

"*I am so grateful to all my colleagues, every time, every day, every minute, always chatting with me and reminding me to be strong and we can make it, we will fight together. All of them are here for me, supporting me ...even if we don't have family here, at least your colleagues, your friends are really there to give support.*"

In comparison, expatriates who lived in Saudi Arabia away from their families faced greater hardship due to their need for support and care. However, under such conditions, some colleagues became close friends and cared more about the infected. Some foreigners were surprised by the support they received from colleagues.

"*Since the start of my home isolation, some of my friends started checking up on me by calling on the phone. Two of them were constantly asking me if I need anything.*

*Every day, when I wake up, they have messages asking how have I been. And they were bringing cooked food and fruits to my doorstep. I never expected that they would care a lot since they were not my closest friends. I was so thankful because their kindness helped me feel better*". An advice from her was "*To my fellow expat who had the virus and living alone, they should be strong because no one can help them during the time of isolation but themselves. Don't forget to pray because God is the great healer. And remember, it's during the worst storms of your life that you will get to see the true colors of the people around you.*"

"*All I needed is the support, I mean if they told me what you will choose when you are dealing with COVID19 patients. I wouldn't give them anything but support! That's all they need. They don't need anything else.*"

Nearly all participants agreed that mental peace and lack of stress are major factors in overcoming illness.

*"Relying on what had happened to me and the patient that I used to see. . . The most important thing in being infected with COVID-19 is the mental health".*

*"You have to be relaxed, you don't have anything negative around you. I mean look at me what made me stronger and able to provide strength to my family is that I've never thought of anything negative."*

*"I really believe that it will start from stress. Once you feel stressed, you will lose your appetite to eat, then all you have to think about is the virus within you and that will make you weak. You need to ignore negativity and advise yourself to be strong. If you cannot eat, you have to force yourself to eat because our bodies need food to energize". "When you feel stress, you will not have energy, the virus will defeat you until you become critical."*

Participants also spoke about their experience with another patient sharing a room during quarantine, whether in a hospital or a hotel. Some patients preferred to be alone rather than have a depressed patient with them, which they felt affected their mental well-being, caused them more stress, and ultimately prolonged their recovery time.

*"It's really better if you are alone in the room than to have someone who will pull you down like the one that I was in that room. She really made me scared".*

Another participant shared the experience of their parents.

*"The mental health and those who talk to him telling him what happened to them that one stayed almost for 10 days in the ICU or 15 days, so he assumed that the same thing will happen to him."*

## Work resignation

The stress from the infection and the negative attitudes from colleagues have pushed some expatriates to leave their jobs. One participant faced pressure from colleagues to return to work immediately after a negative test result, although the participant was still physically and psychologically weak. As a result, signs, and symptoms worsened.

*"I really felt so down and I couldn't stop crying. . .I felt really bad because my close friends who I was expecting to support me during difficult times were the ones adding insult to injury. . . I spent the day thinking a lot, and it made me cry even more. At night, I had a difficulty of breathing and chest heaviness. I tried deep breathing exercises but it didn't get any better. I called the "kind" colleague. She arranged for an ambulance and I was rushed to the COVID-ER. . . The doctor told me it was an anxiety attack. He advised me to avoid stressors."*

*"I was really pushing myself to get better but what my close friends did really affected me. I became more emotional. I was so depressed. Instead of getting better, I became weaker. And at that moment, I decided that I will listen to my mother. I decided it's time for me to resign."*

While family members of the Saudi participants were at close distance, a double burden fell on the expatriates. Thus, some expatriates who lived away from their families went through

difficult experiences when they had to make serious decisions about leaving their jobs and returning to their countries.

> "*As soon as I finished explaining, my mom asked me to resign immediately*". She said I've been working here for how many years already and this is my wake-up call. She didn't even want me to go back to work. She wanted me to go home as soon as possible. She was so worried that she wanted me to book a flight and go home immediately. I explained to her that at this time of the pandemic, it's not easy to just pack up my things and leave. In this difficult situation, health professionals are needed to fight this deadly disease. And even though we want to go home and stay with our families, vacations are still on hold and international flights are not yet open."

## Discussion

A number of studies have examined the experience of healthcare professionals in caring for patients with COVID-19. However, only a few studies have been conducted to examine the experience of healthcare providers as patients. Since the discovery of the first case of COVID-19 in Saudi Arabia and around the world, treatment protocols in hospitals in all countries have focused on treating the physical symptoms of COVID-19, often overlooking the psychological impact. This study examined the life experiences of Saudi and foreign healthcare providers who work in hospitals in the eastern region of Saudi Arabia and have recovered from COVID-19.

In this study, one of the main aspects of the patient's experiences encountered was the psychological impact of the disease, which often impaired their physical health and recovery. Thus, while the participants consciously chose to be isolated, the consequences worsened their mental health, including fear, constant feelings of loneliness, and being away from home. These stressors had a profound effect even when returning to normal life due to fear of being a transmitter of disease, consistent with studies conducted among the general public [21, 22]. Moreover, loneliness due to family absence is considered one of the main risk factors for depression and anxiety in patients with COVID-19 [23, 24].

The fears of the participants in this study about returning to normal life were not only associated with the transmission of disease but included a whole range of factors that, in some cases, led to depression and resignation. For example, fear of serious complications and possible death, leaving the children at home alone, and worrying about what might happen and who would take care of them were important stressors. In addition, some participants were worried about how to break the news to their children, siblings, and family members, especially the elderly. The latter was related not only to the fear of disease transmission but also to how society, including family and friends, would perceive this news. This forced the participants to hide their pain from others and pretend they could easily tolerate the disease. As a result, despite the constant need for emotional support to reduce stress and look after their own well-being, participants in this and earlier studies had to take on increased responsibility for comforting others and demonstrate or pretend that they were doing well and stable [25–27].

One of the main findings of this study was the negative stigma toward patients with COVID-19. This attitude was criticized by the participants, who were surprised by the reaction and felt stigmatized and shamed. It was expected to find fear and anxiety among non-specialists, but surprisingly, similar responses were found among many healthcare professionals with medical backgrounds. Participants felt that social media played an important role in shaping

this perception, especially during the first few months of uncertainty about the severity of the infection and its complications [24]. This fear has led people to believe that COVID-19 is a deadly infection and has led some to flee isolation or hide their travel history to protect themselves. As a result, even those who voluntarily went into isolation were treated most disrespectfully and intolerantly [28, 29].

It is well-known that healthcare providers are the cornerstone of the health system, regardless of their position and status. Therefore, to provide good care and support to patients during and after pandemics, efforts must be made to ensure that they are in good physical and mental health. However, the experience of the past two years and this study showed that, in addition to the workload and caring for others, healthcare professionals experienced personal drama and stress that will have a long-term impact on their future work and communication with family and colleagues. All of the negative physical and mental signs reported in other studies, such as exhaustion, fear, anxiety, and sadness, can affect patients who end up feeling more tense, desperate, and hopeless. Such feelings can affect the immune system and make patients feel worse [21, 30, 31].

This stigma can lead to losing trust in the employer, colleagues, and family. Therefore, it is recommended that future research explore the role of social media and society in both positive and negative impacts on the health of a patient or healthcare provider as a patient. Furthermore, hospital leaders and policymakers must invest their efforts and hospital resources in improving these aspects, which can improve the physical and mental well-being of healthcare providers.

Further data from this study showed that foreign workers experienced an even greater burden since they were away from family and had no support other than occasional support from colleagues. Similarly, data from previous studies have shown that the experiences of COVID-19 survivors in the intensive care unit were dominated by themes of fear of intubation, death alone or away from family, feeling insecure about their families if they died, wishing they could die near close ones [3, 21].

When mental health support was mentioned, participants tended to talk more about employer support, pushing participants to return to work immediately after a negative test result, not allowing enough time for a full recovery. As a result, some foreign workers had to resign and leave. Given the lessons learned from COVID-19, the shortage of healthcare providers in times of crisis, and the importance of how employers relate to their staff, future research with this population could help focus on what support they should have received and what might make them stay or leave. In addition, the healthcare system and hospitals should consider standard support for healthcare professionals in the long term, as a worker who has not fully recovered or has an emotional disability cannot be a strong asset to the team [13, 32]. Conversely, a motivated worker who experiences not only occasional care from colleagues but also comprehensive support from the employer can become a mentally healthy workforce for the sustainability of health services.

The findings of the current study have made a significant contribution to the existing literature and future research. First of all, this study showed that healthcare providers, as patients, experience the same difficulties and stressors as the general public. In some cases, these factors are even worse, as a new type of stigma is developed by family members, colleagues, and employers. Secondly, although employers treated foreign employees unfairly, colleagues treated them well. At the same time, some participants reported outright insensibility of colleagues after hearing about a positive test. Therefore, in future research, it is recommended to study the attitude of medical workers toward their colleagues, depending on whether they are local or foreign. Third, this research has created a range of frameworks for designing and implementing educational projects among different levels of society to educate them and likely

influence their attitudes and perceptions during a health crisis. Finally, given the impact of social media and the flow of information of any type, more research is needed to examine the sources used to obtain information by the general public, whether these sources are reliable, and how the public can be taught to use only scientific data and not social data.

## Limitations

In addition to the added value of this study, certain limitations warrant discussion. Given the sensitive nature of the topic under study, future studies may consider a mixed methods approach that will study the allostatic load of healthcare providers, study their daily workload using quantitative approaches, and develop a qualitative phase based on quantitative data. This will create a broader picture of the research topic in a more comprehensive perspective beyond COVID-19. Lastly, the sampling method also did not allow full blinding of the participants, an inherent limitation that needs to be kept to a minimum as much as possible.

## Conclusions

Understanding the experience of healthcare providers as patients during the pandemic has allowed to look at the feelings and needs of people during illness from a new perspective. As expressed by participants, being a healthcare provider does not reduce the fear of the disease and does not mitigate its consequences in the form of stigmatization and isolation. In addition to its added value, this study has set the stage for future similar studies that can explore the social and occupational stigma and challenges faced by healthcare providers when dealing with any health condition, not only COVID-19. This wide-ranging approach to research will allow the development of educational programs and initiatives to increase the knowledge of colleagues and management about the dangers of a given health condition for those who treat patients, regardless of their occupational or demographic characteristics.

## Author Contributions

**Conceptualization:** Abbas Al Mutair, Amal I. Al Hassawi, Mohammed I. Al Bazroun, Fatimah S. Alahmed.

**Data curation:** Abbas Al Mutair, Amal I. Al Hassawi, Mohammed I. Al Bazroun, Fatimah S. Alahmed, Mary A. Defensor.

**Formal analysis:** Abbas Al Mutair, Zainab Ambani.

**Investigation:** Abbas Al Mutair.

**Methodology:** Abbas Al Mutair.

**Project administration:** Abbas Al Mutair.

**Resources:** Abbas Al Mutair.

**Supervision:** Abbas Al Mutair.

**Validation:** Abbas Al Mutair.

**Writing – original draft:** Abbas Al Mutair, Faiza Aljarameez.

**Writing – review & editing:** Abbas Al Mutair, Alexander Woodman, Chandni Saha.

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
