## [Decision Letter · Decision Letter 0]

25 Jun 2023

PONE-D-22-34667Healthcare Providers as COVID-19 Patients: Long-Term Psychological Consequences of Short-Term Physiological ExperiencePLOS ONE

Dear Dr. Al Mutair,

Thank you for submitting your manuscript to PLOS ONE. After careful consideration, we feel that it has merit but does not fully meet PLOS ONE’s publication criteria as it currently stands. Therefore, we invite you to submit a revised version of the manuscript that addresses the points raised during the review process.

We look forward to receiving your revised manuscript.

Kind regards,

Taofiki Ajao Sunmonu

Academic Editor

PLOS ONE

Additional Editor Comments:

The authors have tried to explore the physiological and psychological impact s of Covid 19 on the healthcare workers. This is an important clinical and occupational issues which would throw more lights on the well being of health care providers during the Covid 19 pandemics. tHE Authors need to address the concerns of the eminent reviewers on this article to improve its quality

Reviewers' comments:

Reviewer's Responses to Questions

**Comments to the Author**

1. Is the manuscript technically sound, and do the data support the conclusions?

Reviewer #1: No

Reviewer #2: Yes

2. Has the statistical analysis been performed appropriately and rigorously? 

Reviewer #1: No

Reviewer #2: Yes

3. Have the authors made all data underlying the findings in their manuscript fully available?

Reviewer #1: Yes

Reviewer #2: Yes

4. Is the manuscript presented in an intelligible fashion and written in standard English?

Reviewer #1: Yes

Reviewer #2: Yes

5. Review Comments to the Author

Reviewer #1: The premise of the research is the fact that, in spite of an abundance of research on the clinical (or ‘physiologic’) aspects of COVID-19, the authors feel there is paucity of information on the psychological aspects of COVID-19, especially as viewed through the lens of the experiences of the participants, which are health care workers (HCWs).

However, the inclusion criteria is unclear. It seems that the authors intended to interview HCWs that had COVID-19 and recovered. Additionally, it is unclear if and how these infections were confirmed. Furthermore, the sample size is made of 10 participants (8 nurses & 2 food servers), and as such is hardly representative of HCWs.

It is also unclear why the authors chose to use snowball sampling, considering that the sample size was small, and the population of interest was hardly inaccessible.

Reviewer #2: The authors have evaluated the long-time psychological effect of the COVID pandemic on health care workers. This is a pertinent article using the lenses of the lived experiences of healthcare workers to evaluate the impact of the pandemic on their psychological state. I have a few comments.

1. I advise that the authors revise the title of the manuscript. The COVID-19 experience is both physiologic and psychological and some might argue there is a spiritual ramification to it. It’s best to leave it as COVID-19 experience rather than defining it as a physiological one.

2. The list of thematic categorizations should be removed from the body of the manuscript. This could be made into a table.

3. In figure one, please provide the legends to delineate the major themes and subthemes. I seem to have found only 5 themes instead of seven. Please clarify.

4. The limitation of the study should be discussed. The qualitative method of the study precluded determining patients’ allostatic loads which could influence their perception of the experiences. Data on the support system are hereby inferred and not quantified. The sampling method also did not allow for complete blinding of the participants.

6. PLOS authors have the option to publish the peer review history of their article (what does this mean?). If published, this will include your full peer review and any attached files.

Reviewer #1: No

Reviewer #2: No

---

## [Author Response · Author response to Decision Letter 0]

11 Jul 2023

30 June 2023

Dear Editor,

Re: Revised Manuscript ID PONE-D-22-34667

Please find attached our revised manuscript: “Healthcare Providers as COVID-19 Patients: Long-Term Psychological Consequences of Short-Term Physiological Experience. We would like to thank the reviewer for their thoughtful consideration of our manuscript.

Comments Authors’ Response

Comments to the authors

Reviewer #1: No

Reviewer #2: Yes Thank you for this feedback. Conclusions of the paper were refined.

2. Has the statistical analysis been performed appropriately and rigorously?

Reviewer #1: No

Reviewer #2: Yes Thank you for this comment. The analysis of data was edited to allow reproducibility by future researchers. 

Have the authors made all data underlying the findings in their manuscript fully available? 

Reviewer #1: Yes

Reviewer #2: Yes Thank you 

Is the manuscript presented in an intelligible fashion and written in standard English?

Reviewer #1: Yes

Reviewer #2: Yes Thank you 

Reviewer 1

The premise of the research is the fact that, in spite of an abundance of research on the clinical (or ‘physiologic’) aspects of COVID-19, the authors feel there is paucity of information on the psychological aspects of COVID-19, especially as viewed through the lens of the experiences of the participants, which are health care workers (HCWs).

However, the inclusion criteria is unclear. It seems that the authors intended to interview HCWs that had COVID-19 and recovered. 

Additionally, it is unclear if and how these infections were confirmed. 

Furthermore, the sample size is made of 10 participants (8 nurses & 2 food servers), and as such is hardly representative of HCWs. Done

1. Our qualitative study aimed to understand and describe the life experiences, and particularly the psychological consequences, of healthcare providers who have survived COVID-19 as patients. 

Although we intended to focus on the psychological aspect of COVID-19 survivors, our inclusion criteria did not specify the survivors who had psychological effects, because that would minimize the sample size, as we understood from previous experiences that some HCWs don't see themselves, or don't like to present themselves as having psychological issues. Their self-image, self-presentation and their desire to return to their normal level of productivity at work could hinder their willingness to participate in this study and speak freely and transparently. For that, we made our inclusion criteria more general; however, before the interview, we ensure that all participants are aware of the aim of the study and during the interviews, we were conscious about how to concentrate more on the psychological influences and how the participants felt at those difficult moments and whether they continue to experience any kind of psychological issues. Our questions encouraged participants to talk naturally and give details about their experiences and for some participants, the interviews helped them to ventilate and recognize some stressors that irritated them such as how they felt being isolated and stigmatized as a source of fear and harm to others. This actually gave us a chance to capture every nuance of pain, and psychological impact they had or still have at the moment. 

2. All infected cases for HCWs in Saudi Arabia were must to be confirmed by official COVID-19 test at hospital, and to be reported to Ministry of Health. Therefore, confirmation was granted from the beginning. 

3. Yes, as in qualitative studies, we stopped recruiting participants once we reached the saturation, which was achieved with the 10th participant. In qualitative studies, the sample are not expected to be representative of the whole population especially if the participants belong to similar environments, and leadership systems that would shape their experiences. 

It is also unclear why the authors chose to use snowball sampling, considering that the sample size was small, and the population of interest was hardly inaccessible. We mainly used purposive sampling to recruit participants. However, in some hospitals we asked participants to refer others who fit with our inclusion criteria. We revised this statement in the manuscript and modified it to be "Purposive sampling" only since that is the main sampling technique we used. 

Reviewer 2

The authors have evaluated the long-time psychological effect of the COVID pandemic on health care workers. This is a pertinent article using the lenses of the lived experiences of healthcare workers to evaluate the impact of the pandemic on their psychological state. I have a few comments.

1. I advise that the authors revise the title of the manuscript. The COVID-19 experience is both physiologic and psychological and some might argue there is a spiritual ramification to it. It’s best to leave it as COVID-19 experience rather than defining it as a physiological one. Thank you for this comment. The title was revised into “Healthcare Providers as Patients: COVID-19 Experience” 

2. The list of thematic categorizations should be removed from the body of the manuscript. This could be made into a table. Done 

3. In figure one, please provide the legends to delineate the major themes and sub-themes. I seem to have found only 5 themes instead of seven. Please clarify. That is right. The analysis resulted in five themes and four sub-themes only not seven. We corrected this sentence in the result section of the manuscript. 

4. The limitation of the study should be discussed. The qualitative method of the study precluded determining patients’ allostatic loads which could influence their perception of the experiences. Data on the support system are hereby inferred and not quantified. The sampling method also did not allow for complete blinding of the participants Thank you for this comment. Limitations section was added to the manuscript.

Thank you once again for the feedback received, and I hope that this information fulfills all of your requirements. Should you have any queries or require further information please do not hesitate to contact me for clarification.

---

## [Editor Report · Decision Letter 1]

12 Jul 2023

Healthcare Providers as Patients: COVID-19 Experience

PONE-D-22-34667R1

Dear Dr. Al Mutair,

We’re pleased to inform you that your manuscript has been judged scientifically suitable for publication and will be formally accepted for publication once it meets all outstanding technical requirements.

Kind regards,

Taofiki Ajao Sunmonu

Academic Editor

PLOS ONE
---

## [Editor Report · Acceptance letter]

15 Aug 2023

PONE-D-22-34667R1 

Healthcare Providers as Patients: COVID-19 Experience 

Dear Dr. Al Mutair:

I'm pleased to inform you that your manuscript has been deemed suitable for publication in PLOS ONE. Congratulations! Your manuscript is now with our production department. 

Kind regards, 

on behalf of

Dr. Taofiki Ajao Sunmonu 

Academic Editor

PLOS ONE